# Effects of V–N Microalloying on Microstructure and Property in the Welding Heat Affected Zone of Constructional Steel

**Kaiyu Cui** [1,2,3], **Haifeng Yang** [1,2,4], **Shengjie Yao** [2,4,*], **Zhengrong Li** [3], **Guodong Wang** [1,2,4], **Hongyun Zhao** [1,2,4] **and Xinchen Nan** [1,2]

1   State Key Laboratory of Advanced Welding and Joining, Harbin Institute of Technology, Harbin 150001, China; yjycuiky@pzhsteel.com.cn (K.C.); hf_yang@hit.edu.cn (H.Y.); wanggd@neu.edu.cn (G.W.); hy_zhao66@163.com (H.Z.); hitwh170820317@163.com (X.N.)
2   Shandong Provincial Key Laboratory of Special Welding Technology, Harbin Institute of Technology (Weihai), Weihai 264209, China
3   State Key Laboratory of Vanadium and Titanium Resources Comprehensive Utilization, Pangang Group, Panzhihua 617000, China; yjjylzr@pzhsteel.com.cn
4   School of Materials Science and Engineering, Harbin Institute of Technology (Weihai), Weihai 264209, China
*   Correspondence: shj_yao2010@hit.edu.cn; Tel.: +86-631-5687-324; Fax: +86-631-5687-305

**Abstract:** Shielded metal arc welding and welding thermal simulation experiment were carried out for constructional steel containing 0% V and 0.10% V, and the microstructure, precipitation feature, microhardness HV0.2, and −20 °C impact value in the welding heat affected zone (HAZ) were investigated. The results showed that in the coarse-grained heat affected zone (CGHAZ), V and N were completely dissolved in the matrix of steel containing 0.10% V to promote the growth of prior austenite grains, meanwhile the fraction of high angle grain boundaries (HAGBs) decreased, thereby leading to the mean −20 °C impact value decreases from 87 J to 18 J. In the grain refined heat affected zone (GRHAZ), V(C, N) precipitates experience re-dissolution and re-precipitation at grain boundaries, V–N microalloying changes the microstructure from lath bainite + granular bainite + small amount of polygonal ferrite to polygonal ferrite + pearlite + martensite, thereby leading to the mean microhardness decreases from 335 HV0.2 to 207 HV0.2, and the mean −20 °C impact value decreased from 117 J to 103 J. In the intercritical heat affected zone (ICHAZ), V(C, N) precipitates experience re-dissolution, re-precipitation, and growth, causing the formation of micro-sized V(C, N) precipitates, thereby leading to the mean −20 °C impact value decreases from 93 J to 62 J.

**Keywords:** vanadium; nitrogen; constructional steel; HAZ; microstructure; hardness; toughness





## 1. Introduction

Steel structure construction possesses advantages such as light weight, high strength, convenient installation, short construction period, excellent earthquake resistance, and high recycling rate [1–3]. In light of microstructural transformation in the weld joint, especially the heat affected zone, brittle or a soft zone usually occurs. Thus, the safety and service performance of steel structural construction not only depends on the property of steel, but also greatly depends on the property of the weld joint [4–6]. In order to achieve excellent properties such as high strength and toughness, a certain content of alloy elements are generally added. The V–N micro-allying technique can realize precipitation strengthening and refinement strengthening through precipitation and the Zener pinning effect on grain boundary of V carbonitride, thereby significantly increasing the strength and toughness of steels [7–12].

However, the dissolution and precipitation of V carbonitride in the weld thermal cycle will affect the microstructure and property of the weld joint. Lei Y. et al. [13] carried out weld experiment on martensitic steel containing various V contents, and found that by increasing the addition of V, the amount of V carbide increased to increase the hardness

and strength level of steels while decreasing the impact toughness. Similar research results were obtained by Sun J. et al. [14], indicating that by increasing the V content to 0.18%, the volume fraction of MC carbide increased, leading to improved strength while maintaining good ductility. However, in the process of post-weld heat treatment, a high fraction of nanoscale MC carbides inhibited the dislocation movement, thereby deteriorating the toughness. In the weld joint, the heat affected zone (HAZ) is quite critical, which may experience microstructural transformation such as grain coarsening to deteriorate the weld joint performance. Chen Y. T. et al. [15] investigated the microstructure and property of the HAZ of high strength low alloy steels containing 0%, 0.047%, 0.097%, or 0.151% V; as the V content increases, the size of the austenite, ferrite, and M–A constituent increases in the HAZ, thus improving the strength of the steel but decreasing the impact toughness. For the V–N microalloying steel, V carbonitride could not precipitate due to a high cooling rate at the coarse-grained heat affected zone (CGHAZ), so the austenitic grains easily became coarsened [16]. Hutchinson B. et al. [17] compared the microstructure and property in the HAZ of the V–N microalloyed steel and Nb microalloyed steel, and indicated that with the decrease in cooling rate, an increased transformation temperature in the HAZ of the V–N microalloyed steel led to the formation of networks of coarse ferrite grains along the prior austenite grain boundaries, thereby resulting in the increases in the impact transition temperature. However, Shi Z. et al. [18] compared the V–Ti microalloying steel and the V–N–Ti microalloying steel, and showed that an increase in N content not only refined the austenitic grains, but also promoted the intergranular ferrite nucleation, therefore, a better combination of strength and toughness was expected in the CGHAZ of the V–N–Ti microalloying steel. Fan H. et al. [19] investigated the microstructure and property in CGHAZ of a low carbon Mo–V–Ti–B steel, and indicated that when the N content increased from 0.0085 wt.% to 0.0144 wt.%, the number of particles increased and the coarse particles with more V(C, N) on the exterior could effectively promote the nucleation of intragranular acicular ferrite and intergranular ferrite, resulting in a finer microstructure, thereby increasing the impact toughness of the CGHAZ. Zhang J. et al. [20] indicated that high-N (240 ppm) V alloyed steel could achieve good weldability because of the formation of V(C, N) precipitates in the austenite region, which act as heterogeneous nucleation sites of intragranular polygonal ferrite and acicular ferrite, but V(C, N) precipitates coarsen with the increase in cooling time $t_{8/5}$ to reduce the impact toughness in CGHAZ.

According to the above-mentioned preliminary studies, weld thermal cycle can give rise to changes in the fraction of precipitates and the size of the grains. This will eventually lead to a local soft or brittle zone at the HAZ, thereby easily causing the failure of steel structure construction at a large load, especially in earthquake conditions. Therefore, the present paper carried out shielded metal arc welding and welding thermal simulation experiments on V–N microalloying constructional steel to investigate the effects of V and N on the microstructure and property of the HAZ of construction steel.

## 2. Materials and Methods

### 2.1. Materials

Experimental materials were smelted by a VAR-150 consumable electrode vacuum furnace (Baotai Equipment Technology Co. Ltd., Baoji, China) and rolled from 70 mm into a 6 mm thick steel plate; the finish temperature and coiling temperature were 860 °C and 650 °C, respectively; and the chemical composition and carbon equivalent values of the tested steels is shown in Table 1. The 0 V tested steel contained 0% V and 0.0021% N, while the 10 V steel contained 0.10% V and 0.0155% N. In order to realize good weathering resistance, certain contents of Cu, Ni, and Cr were added into both tested steels. According to Equation (1), which is suggested by the International Institute of Welding (IIW), the carbon equivalent values (CE) of the tested steels were calculated, and the weldability of the tested steels was poor due to high CE.

$$CE = C + \frac{Mn}{6} + \frac{Cr + Mo + V}{5} + \frac{Ni + Cu}{15} \qquad (1)$$

where C, Mn, Cr, Mo, V, Ni, and Cu are weight percentages of the elements C, Mn, Cr, Mo, V, Ni, and Cu in steel, respectively.

**Table 1.** Chemical composition and carbon equivalent values of tested steels (wt.%).

| Steel | C | Si | Mn | P | S | Cu | Ni | Cr | V | Als | N | CE |
|-------|------|------|------|-------|-------|------|------|------|------|-------|--------|------|
| 0 V | 0.07 | 0.39 | 1.45 | 0.011 | 0.002 | 0.36 | 0.31 | 0.63 | - | 0.036 | 0.0021 | 0.48 |
| 10 V | 0.07 | 0.39 | 1.44 | 0.010 | 0.002 | 0.35 | 0.31 | 0.63 | 0.10 | 0.034 | 0.0155 | 0.50 |

*2.2. Methods*

2.2.1. Shielded Metal Arc Welding Experiment

Weld joint samples were prepared by shielded metal arc welding using a WS-300 DC welding power source (Ruiling Industry Co. Ltd., Shenzhen, China) and J857CrNi high strength welding rod, and the chemical composition of the J857CrNi deposited metal is shown in Table 2. Referring to the welding parameters used in real application of tested steels, a unilateral 30° V shape groove was processed at the tested steel plate, and the welding current was 160 A.

**Table 2.** Chemical composition of J857CrNi deposited metal (wt.%).

| Elements | C | Si | Mn | Ni | Cr | Mo |
|----------|--------|--------|--------|--------|--------|--------|
| Reference content | ≤0.10 | ≤0.80 | ≥0.50 | ≥1.75 | ≥0.30 | ≥0.20 |

2.2.2. Welding Thermal Simulation Experiment

The HAZ is the most important zone that affects the microstructure and property of the weld joint, but the various zones in the HAZ are too narrow to obtain available samples, therefore, a welding thermal simulation experiment was carried out, and various zones in the real weld HAZ were analyzed through the microstructure and property characteristics of welding thermal simulation samples. Welding thermal simulation samples were cut from tested steels and processed into 5 mm × 10 mm × 55 mm sizes, and the welding thermal simulation experiment was carried out by a Gleeble 3500 (DSI, Saint Paul, MS, USA) with various peak temperatures $T_p$ and cooling times $t_{8/5}$ (time for cooling from 800 °C to 500 °C) to simulate the weld thermal cycle experienced by CGHAZ, the grain refining heat affected zone (GRHAZ), and intercritical heat affected zone (ICHAZ) in real welding process, and the experimental parameters and scheme are shown in Table 3 and Figure 1, respectively.

**Table 3.** Experiment parameters of weld simulation.

| CGHAZ | | GRHAZ | | ICHAZ | |
|-------|-------|-------|-------|-------|-------|
| $T_p$ (°C) | $t_{8/5}$ (s) | $T_p$ (°C) | $t_{8/5}$ (s) | $T_p$ (°C) | $t_{8/5}$ (s) |
| 1300 | 12 | 1000 | 13 | 830 | 17 |

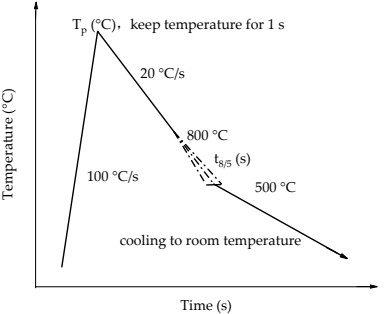

**Figure 1.** Experimental scheme of the weld simulation.

### 2.2.3. Microstructural Characterization and Mechanical Testing

Metallographic samples were prepared and polished, then etched in 4% Nital solution for 15 s, and the microstructure was examined by a DSX510 optical microscope (OM) (Olymbus, Tokyo, Japan). Thin foils were cut from 10 V steel and mechanically ground to 50–60 μm thick slices, then the slices were twin jet electropolished with 5% perchloric acid alcohol solution at −30 °C and subsequently thinned by ion beam. Thin foils were examined by a JEM 2100 transmission electron microscope (TEM) (JEOL, Tokyo, Japan) equipped with an EDS system (JEOL, Tokyo, Japan) to analyze the morphology and distribution of precipitates in 10 V steel. Moreover, the samples of carbon extraction replicas were processed and examined by JEM 2100 TEM, according to the McCall–Boyd method, sufficient random fields were selected to count the size and volume fraction of precipitates by the following equation:

$$V_f = \frac{1.4\pi}{6} \times \frac{ND_{mean}{}^3}{V}$$

(2)

where $V_f$, $D_{mean}$, N, and V are the volume fraction, mean diameter, quantity, and total volume of precipitates, respectively.

The HV0.2 microhardness of the weld joints was measured by an FM-700 microhardness testing machine (FUTURE-TECH, Kawasaki, Japan) according to GB/T 4340.1-2009, where the test points were perpendicular to the weld seam with a 0.25 mm interval, the test load was 1.96 N, and the load time was 10 s. The Charpy V notch impact samples were cut from the HAZ with a 5 mm distance to the center of the weld seam, processed into 5 mm × 10 mm × 55 mm sizes, and then the −20 °C impact values was measured thrice for every tested steel by the PIT452D-4 impact test machine a (Wance Test Equipment Co. Ltd., Shenzhen, China)ccording to GB/T 229-2020, and the morphology of impact fracture was observed by a MERLIN Compact scanning electron microscope (SEM) (ZEISS, Oberkochen, Germany). Tensile testing was carried out by an Instron 5967 30 KN universal testing machine (Instron, High Wycombe, UK).

## 3. Results

### 3.1. Microstructure and Mechanical Property of Tested Steels

The detailed microstructure and mechanical property of the tested steels are shown in Figure 2 and Table 4, and the mechanical properties were measured thrice for every tested steel. The metallographic structure consisted of ferrite and pearlite, but the grain of 10 V steel was relatively more refined. It is known that V is a strong carbonitride forming element, so that V(C, N) precipitates in the austenite and ferrite during hot rolling, giving rise to grain refinement strengthening and precipitation strengthening. Meanwhile, the yield ratio of 10 V steel was controlled lower than 0.85, which is thought to increase the earthquake resistance of steel structure construction.

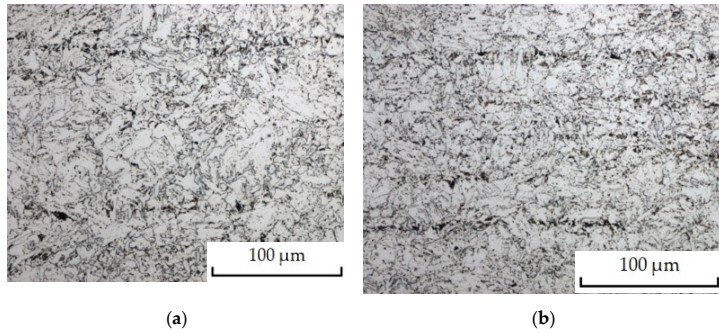

**Figure 2.** Metallographic structure of the tested steels: (**a**) 0 V; (**b**) 10 V.

**Table 4.** Mechanical property of the tested steels.

| Steel | $R_{eL}$ (MPa) | $R_m$ (MPa) | $R_{eL}/R_m$ | A (%) |
|---|---|---|---|---|
| 0 V | 331 ± 8 | 474 ± 13 | 0.70 ± 0.01 | 31.5 ± 1.2 |
| 10 V | 512 ± 12 | 612 ± 11 | 0.84 ± 0.01 | 22.0 ± 1.6 |

The distribution and morphology of V(C, N) precipitates in 10 V steel were examined by TEM in high magnification (20,000×), as shown in Figure 3. The morphology of V(C, N) precipitates was mainly a spherical shape, and these precipitates were observed inside the grains (Figure 3a), nearby grain boundaries (Figure 3b), or dislocation lines (Figure 3c).

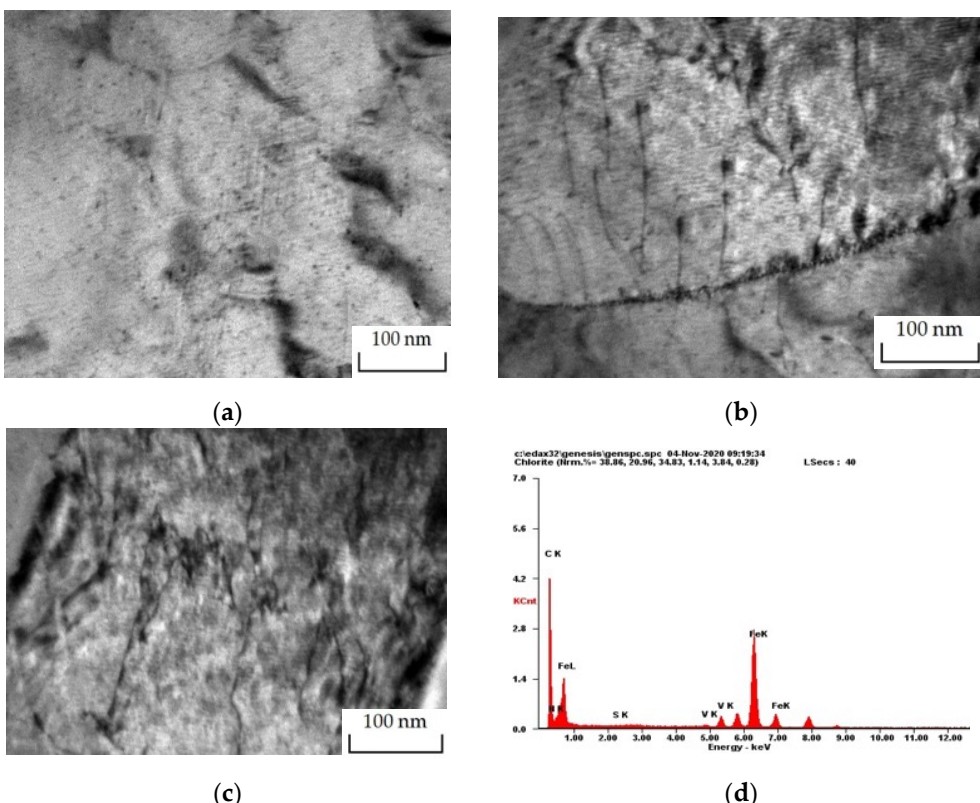

(a)

(b)

(c)

(d)

**Figure 3.** Morphology and distribution of V(C, N) precipitates in the base material of 10 V steel: (**a**) dispersed distribution of V(C, N) precipitates; (**b**) V(C, N) precipitates at the grain boundary; (**c**) V(C, N) precipitates pin dislocation; and (**d**) elemental analysis of precipitate by energy dispersive spectroscopy (EDS).

Quantitative analysis of the volume fraction and size distribution of V(C, N) precipitates in 10 V steel was conducted by TEM with a replica technique. Through the McCall–Boyd method, the volume fraction $V_f$ of V(C, N) precipitates was measured to be approximately 0.196%. The size distribution of V(C, N) precipitates (Figure 4) implies the appearance of two peaks of size distribution. The V(C, N) precipitates with smaller size (2–<18 nm) mainly precipitated in the ferrite grains after coiling, and the volume fraction $V_f$ was 0.145% while the V(C, N) particles with a larger size (18–50 nm) mainly precipitated in the pearlite during hot rolling, and the volume fraction $V_f$ was 0.051%. The peak value of the size distribution of V(C, N) precipitates with a small size was in the range of 2–10 nm, while the peak value of the size distribution of V(C, N) precipitates with a large size was in the range of 18–30 nm.

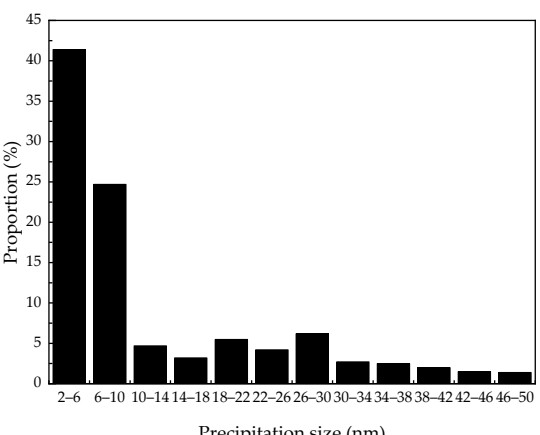

**Figure 4.** Size distribution of V(C, N) precipitates in the base material of the 10 V steel.

### 3.2. Microstructure and Properties of Weld Joint

The tensile property of the weld joints of 0 V and 10 V steels is shown in Table 5. The fracture position of tensile samples of both tested steels was located at the base material, indicating that the welding parameters and welding material matched well, and the strength of the weld joints was higher.

**Table 5.** Mechanical property of the welded joints.

| Steel | $R_{eL}$ (MPa) | $R_m$ (MPa) | A (%) | Fracture Position |
|-------|----------------|-------------|-------|-------------------|
| 0 V   | 326            | 473         | 32.5  | Base material     |
| 10 V  | 494            | 604         | 20.0  | Base material     |

The microhardness distribution of the weld joints of 0 V and 10 V steels is shown in Figure 5. No softening zone existed in the weld joints of both tested steels, which is in accordance with the phenomenon that the tensile samples fractured at the base material. The microhardness of the weld seam of both tested steels was relatively high, which resulted from the high strength of the welding rod. For 0 V steel, the microhardness of GRHAZ was the highest, while ICHAZ was the lowest. For 10 V steel, the microhardness of CGHAZ was the highest, while ICHAZ was the lowest. Compared to the 0 V steel, the microhardness of the CGHAZ of 10 V steel was higher, while the microhardness of GRHAZ and ICHAZ of the 10 V steel was lower, so it can be concluded that the addition of 0.10% V and 0.0155% N in steel will affect the microhardness of HAZ.

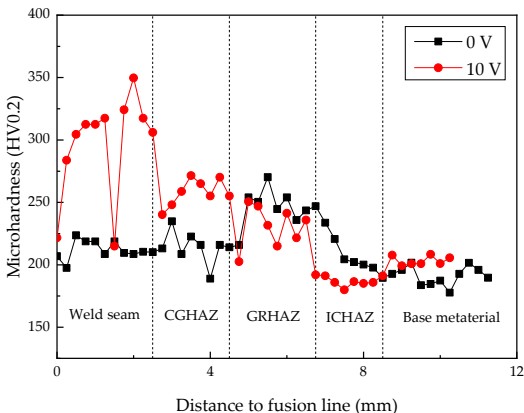

**Figure 5.** Microhardness distribution of the weld joints.

The impact values of HAZ of 0 V and 10 V steels are shown in Table 6. Compared to the 0 V steel, the impact value of HAZ of 10 V steel significantly decreased at −20 °C, and the mean value decreased by approximately 40 J.

**Table 6.** Low temperature impact value of real weld HAZ.

| Steel | −20 °C Impact Value (J) | | | |
|---|---|---|---|---|
|  | Value 1 | Value 2 | Value 3 | Mean Value |
| 0 V | 101 | 105 | 109 | 105 |
| 10 V | 57 | 85 | 57 | 65 |

The typical impact fracture morphologies of the HAZ of both tested steels (Figure 6) showed that dimples were observed in the impact fracture of the HAZ of both tested steels. Micro holes caused by plastic deformation experience nucleation, growth, and aggregation to generate dimples. Therefore, the impact fracture of both steels was characterized by ductile fracture. In comparison, the dimples were smaller and shallower for 10 V steel, and some cleavage planes were observed, therefore the impact toughness of the HAZ of 10 V steel was relatively low. Therefore, it can be concluded that the addition of 0.10% V and 0.0155% N in steel will make the fracture mode of impact samples of a real weld HAZ transform from ductile fracture to quasi-cleavage fracture, corresponding to the decreased impact toughness.

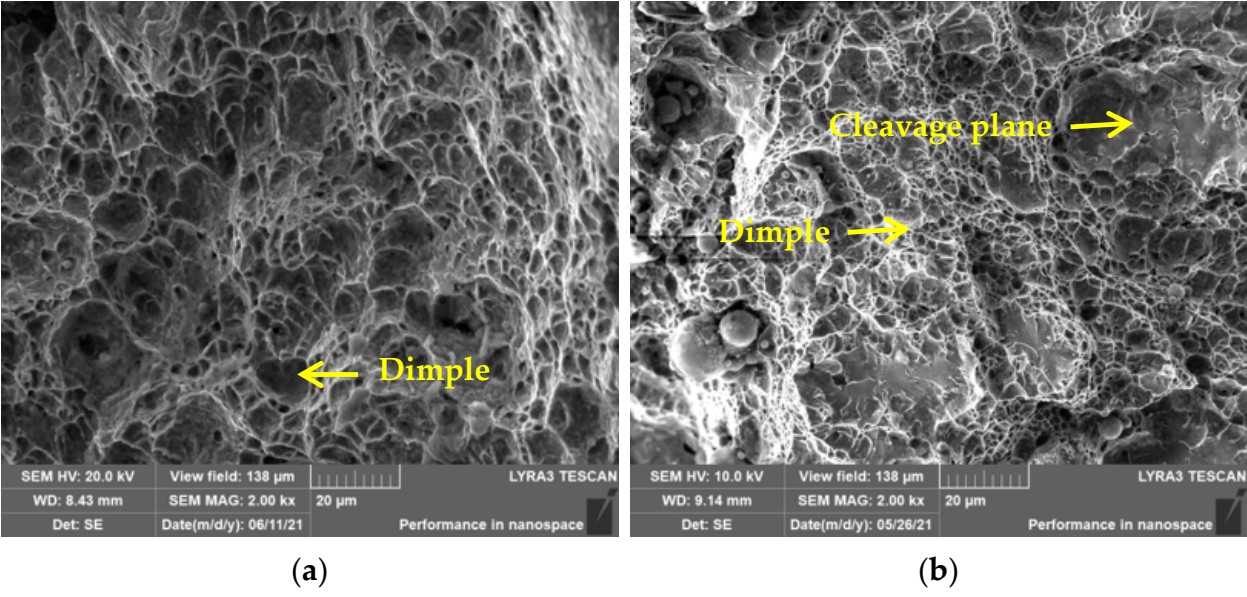

(**a**)　　　　　　　　　　　　　　　　　　　　　(**b**)

**Figure 6.** Morphology of the impact fracture of real weld HAZ: (**a**) 0 V; (**b**) 10 V.

The metallographic structure of weld joints of the tested steels is shown in Figure 7, and there was no obvious difference between the 0 V and 10 V steels. The metallographic structure of CGHAZ consisted of lath bainite, granular bainite, and a small amount of polygonal ferrite, and it can be observed that the M–A constituent in an irregular block shape was uniformly distributed in the ferrite grains. The metallographic structure of GRHAZ consisted of polygonal ferrite, pearlite, and martensite, and the grain size of both tested steels was almost consistent. The metallographic structure of ICHAZ consisted of polygonal ferrite, pearlite, and a small amount of martensite, and the grains of the 10 V steel were relatively refined.

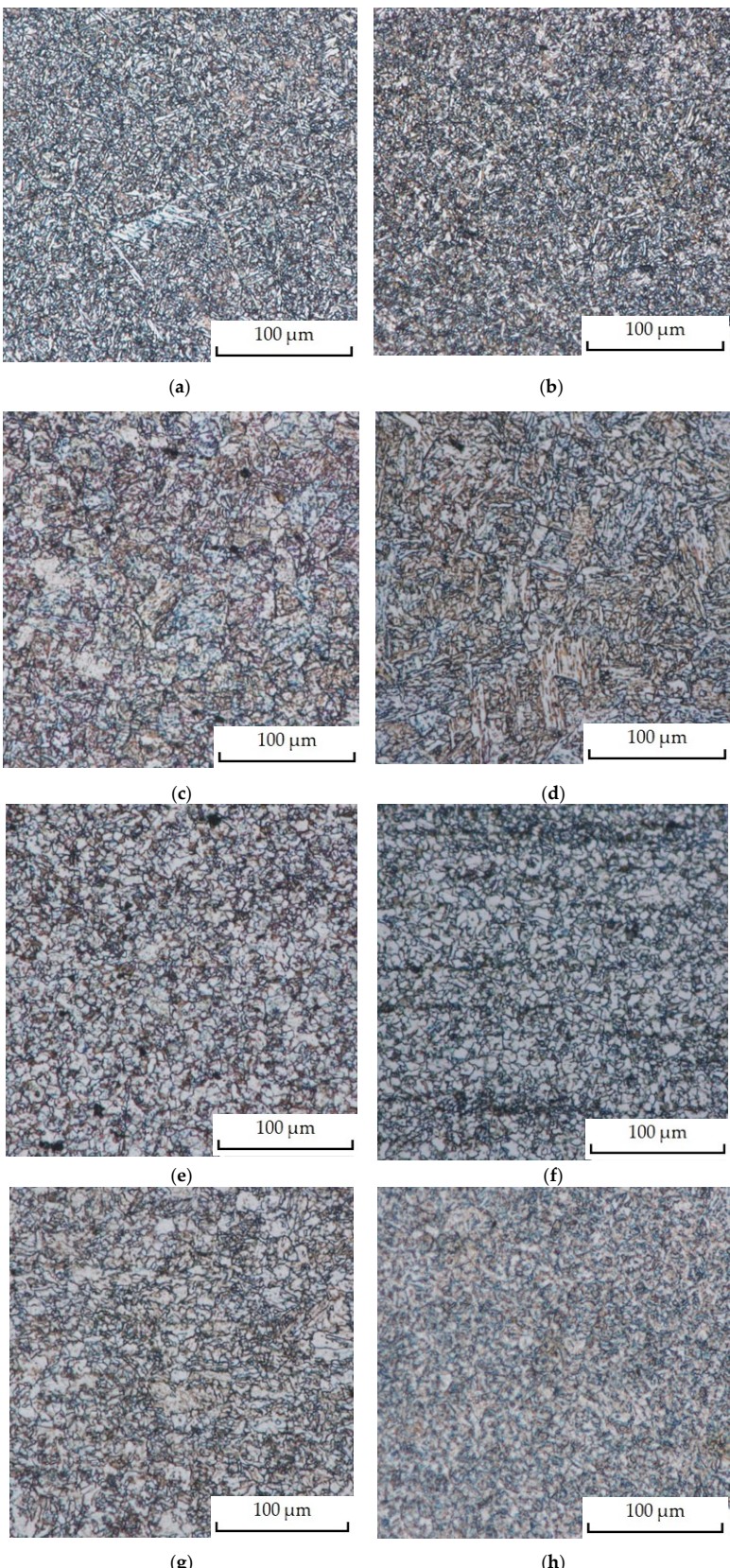

**Figure 7.** Metallographic structure of the weld joints: (**a**) fusion zone of 0 V; (**b**) fusion zone of 10 V; (**c**) CGHAZ of 0 V; (**d**) CGHAZ of 10 V; (**e**) GRHAZ of 0 V; (**f**) GRHAZ of 10 V; (**g**) ICHAZ of 0 V; and (**h**) ICHAZ of 10 V.

### 3.3. Microstructure and Property of Welding Thermal Simulation Samples

The microhardness of the welding thermal simulation samples of 0 V and 10 V steels is shown in Figure 8a. For the 0 V steel, the microhardness of GRHAZ was the highest, while ICHAZ was the lowest. For 10 V steel, the microhardness of CGHAZ was the highest, while GRHAZ was the lowest. Compared to the 0 V steel, the microhardness of the simulated HAZ of 10 V steel was lower, especially for GRHAZ, the mean microhardness decreased from 335 HV0.2 to 207 HV0.2. Similarly, the −20 °C impact test was carried out on the welding thermal simulation samples of 0 V and 10 V steels, and the impact values are shown in Figure 8b. For both tested steels, the impact value of simulated GRHAZ was the highest, while CGHAZ was the lowest. Compared to 0 V steel, the impact value of the simulated HAZ of 10 V steel was lower, and the mean −20 °C impact values of CGHAZ, GRHAZ, and ICHAZ decreased from 87 J, 117 J, 93 J to 18 J, 103 J, 62 J, respectively. Therefore, it can be concluded that the addition of 0.10% V and 0.0155% N in steel will increase the strength while decreasing the low temperature impact toughness of the HAZ.

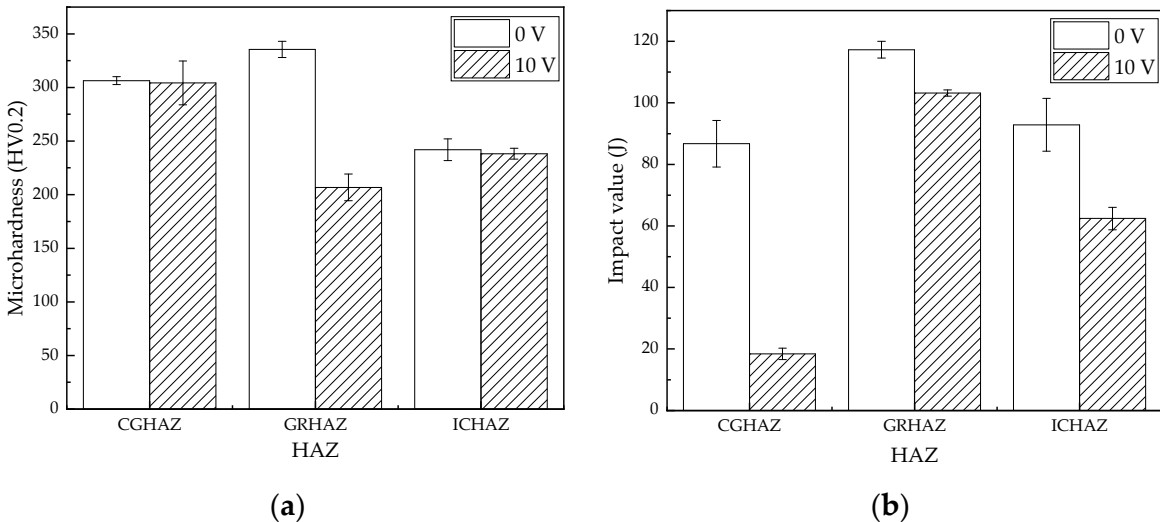

**(a)**                    **(b)**

**Figure 8.** Comparison of the properties of the welding thermal simulation samples: (**a**) microhardness; (**b**) impact value at −20 °C.

The impact fracture morphologies of the welding thermal simulation samples (Figure 9) demonstrate that the fracture of simulated CGHAZ, GRHAZ, and ICHAZ of 0 V steel was characterized by ductile fracture. In contrast, a river pattern was observed in the impact fracture of the simulated CGHAZ of 10 V steel, showing that cleavage cracks propagated on various planes through secondary cleavage or intersection with screw dislocation. Therefore, the impact fracture of the simulated CGHAZ of 10 V steel is characterized by cleavage fracture, namely brittle fracture. Therefore, according to the impact fracture morphology of the welding thermal simulation samples in the −20 °C impact test, it can be concluded that the addition of 0.10% V and 0.0155% N in steel will make the fracture mode of the simulated CGHAZ transform from ductile fracture to cleavage fracture.

The microstructure of the welding thermal simulation samples of 0 V and 10 V steels (Figure 10) showed no obvious difference between the two simulated CGHAZ of 0 V and 10 V steels, mainly consisting of lath bainite, granular bainite, and a small amount of polygonal ferrite. However, the microstructure constitution of the simulated GRHAZ of 0 V and 10 V steels was quite different. The former consisted of lath bainite, granular bainite, and a small amount of polygonal ferrite, while the latter consisted of uniform and refined polygonal ferrite, pearlite, and martensite. For the simulated ICHAZ, both 0 V and 10 V steels exhibited polygonal ferrite, pearlite, and martensite, but the volume fraction of martensite of 0 V steel was larger, and the grain of 10 V steel was refined.

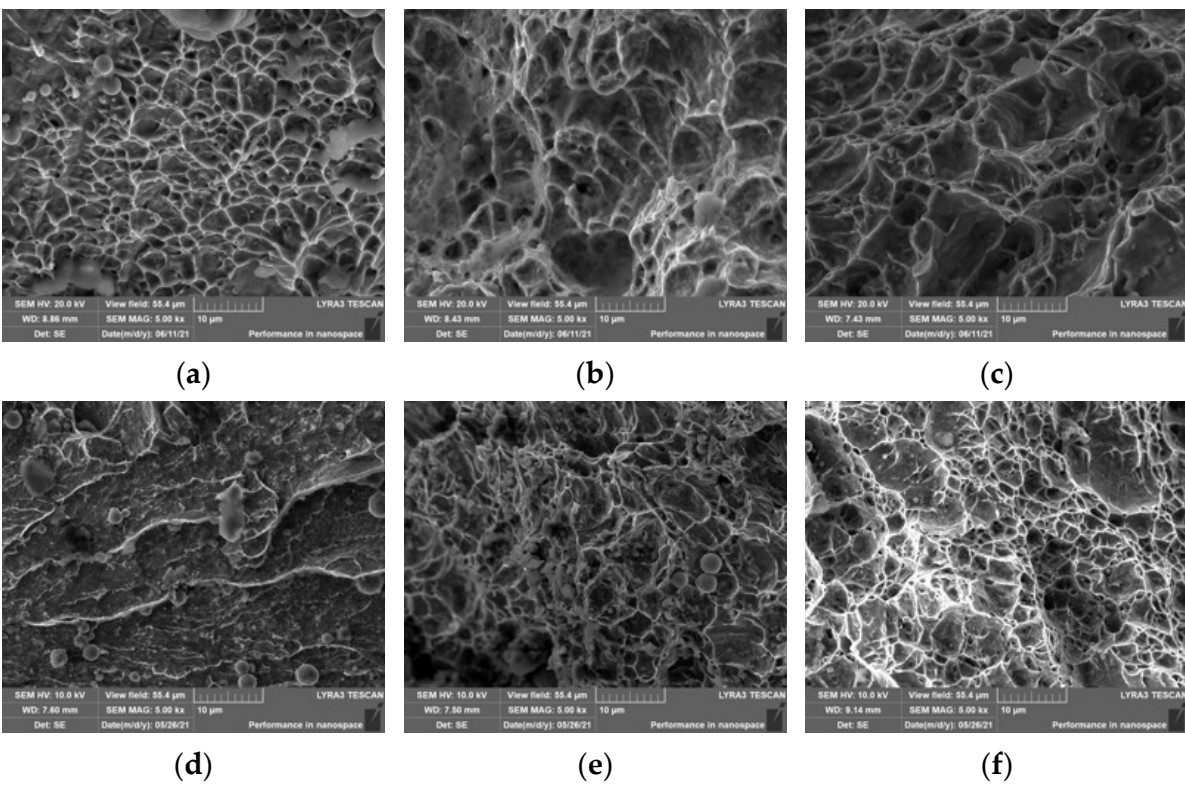

**Figure 9.** Morphology of the impact fracture of the welding thermal simulation samples: (**a**) CGHAZ of 0 V; (**b**) GRHAZ of 0 V; (**c**) ICHAZ of 0 V; (**d**) CGHAZ of 10 V; (**e**) GRHAZ of 10 V; and (**f**) ICHAZ of 10 V.

In order to further analyze the differences in the microstructure of the welding thermal simulation samples between the 0 V and 10 V steels, EBSD was carried out [21–25], and the testing results are shown in Figures 11–13. The IPF maps (Figure 11) of the simulated CGHAZ of the 0 V and 10 V steels showed polygonal and lath prior austenite grains (PAGs), and the PAG size of the simulated CGHAZ of 10 V steel was larger. The simulated GRHAZ of the 0 V steel showed lath PAGs, while the 10 V steel exhibited uniform and refined polygonal ferrite grains. However, both the simulated ICHAZ of 0 V and 10 V steels showed quasi-polygonal ferrite grains.

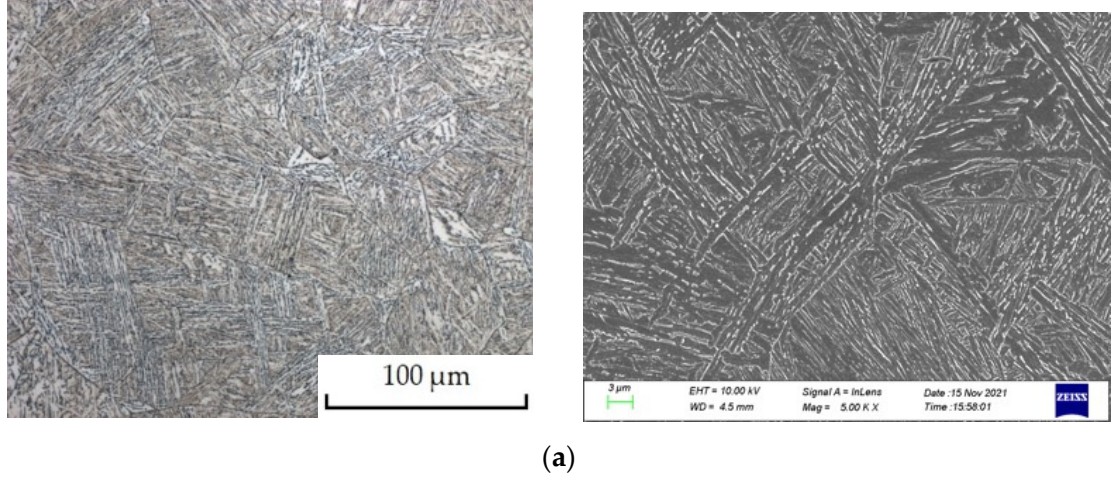

(**a**)

**Figure 10.** *Cont.*

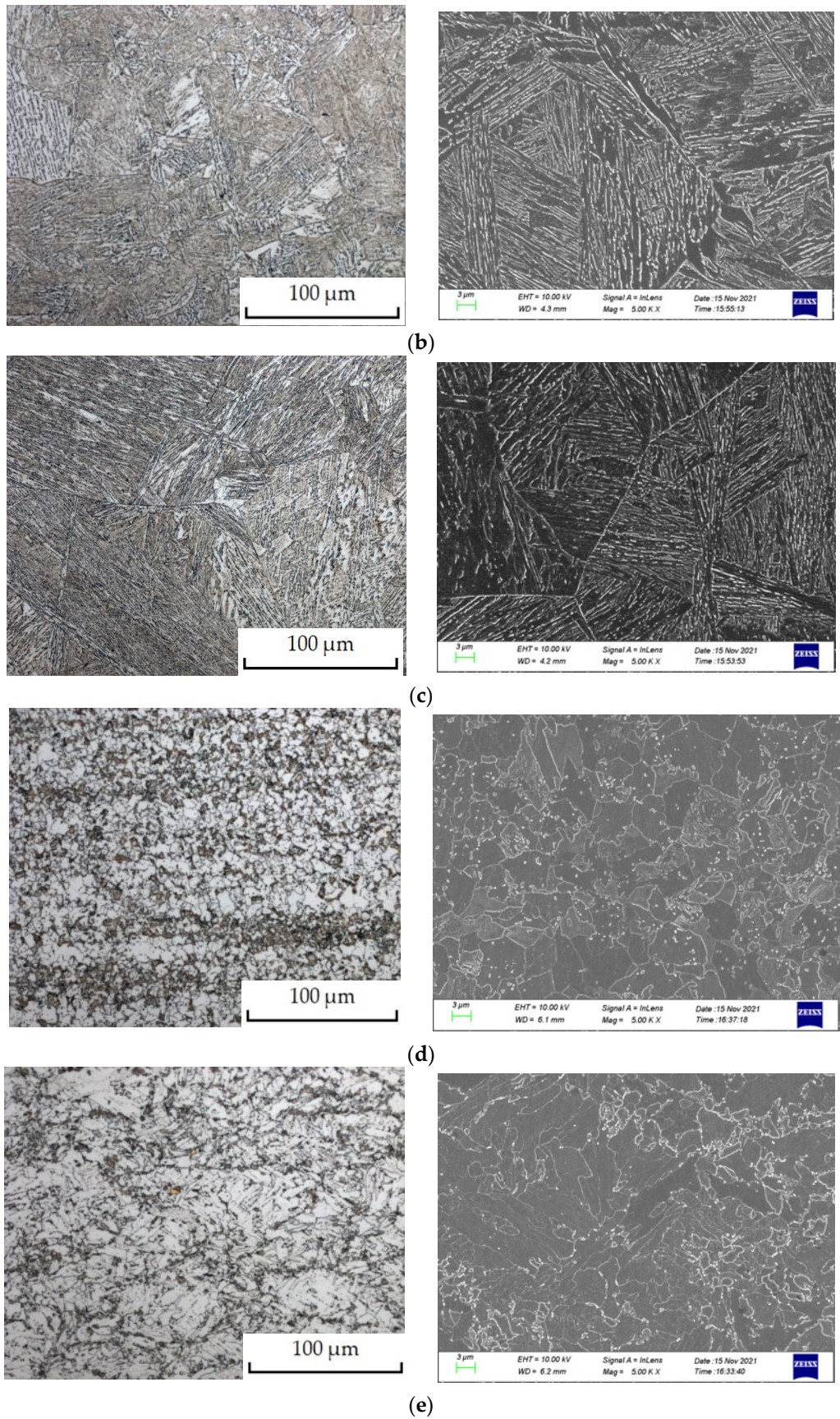

**Figure 10.** *Cont.*

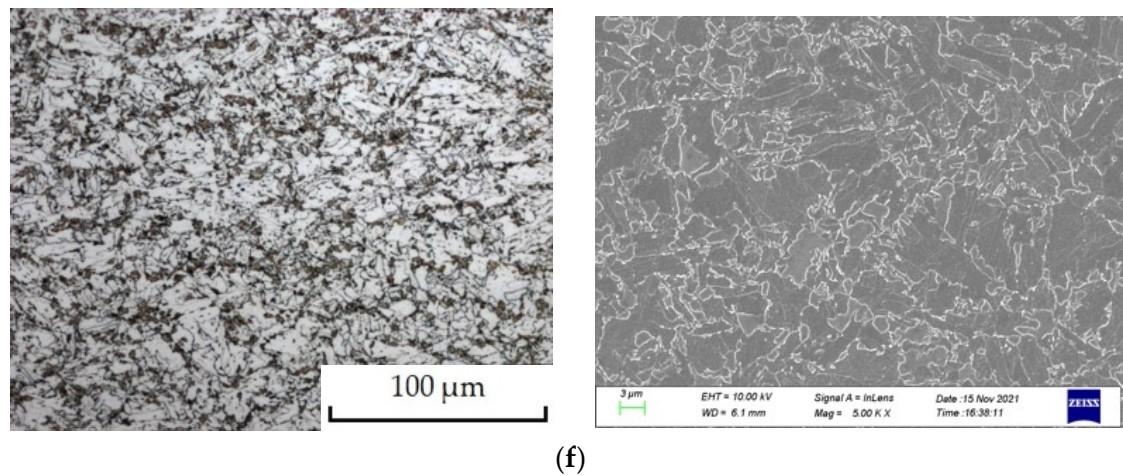

(**f**)

**Figure 10.** Microstructure of the welding thermal simulation samples (left: OM; right: SEM): (**a**) CGHAZ of 0 V; (**b**) CGHAZ of 10 V; (**c**) GRHAZ of 0 V; (**d**) GRHAZ of 10 V; (**e**) ICHAZ of 0 V; and (**f**) ICHAZ of 10 V.

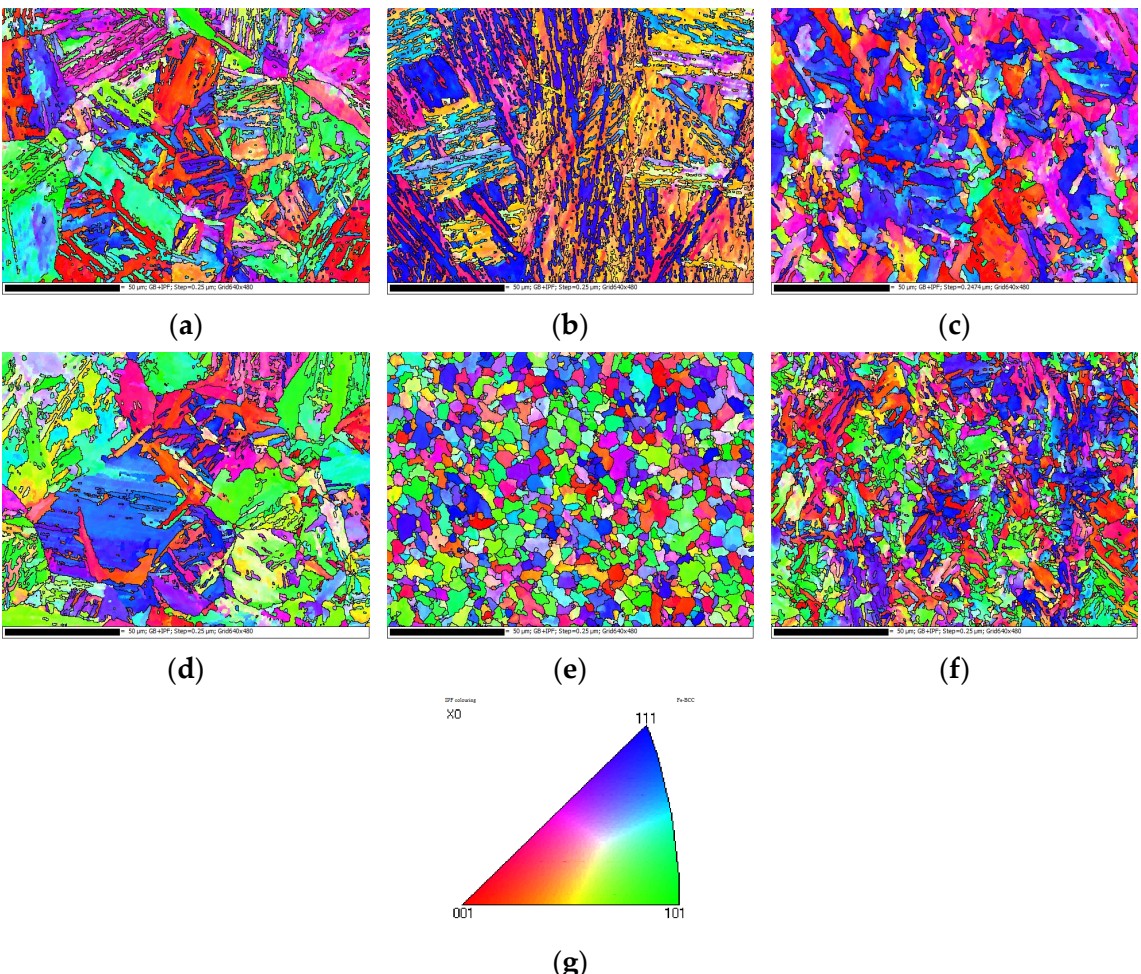

**Figure 11.** IPF maps of the welding thermal simulation samples: (**a**) CGHAZ of 0 V; (**b**) GRHAZ of 0 V; (**c**) ICHAZ of 0 V; (**d**) CGHAZ of 10 V; (**e**) GRHAZ of 10 V; and (**f**) ICHAZ of 10 V; (**g**) crystallographic orientation illustration.

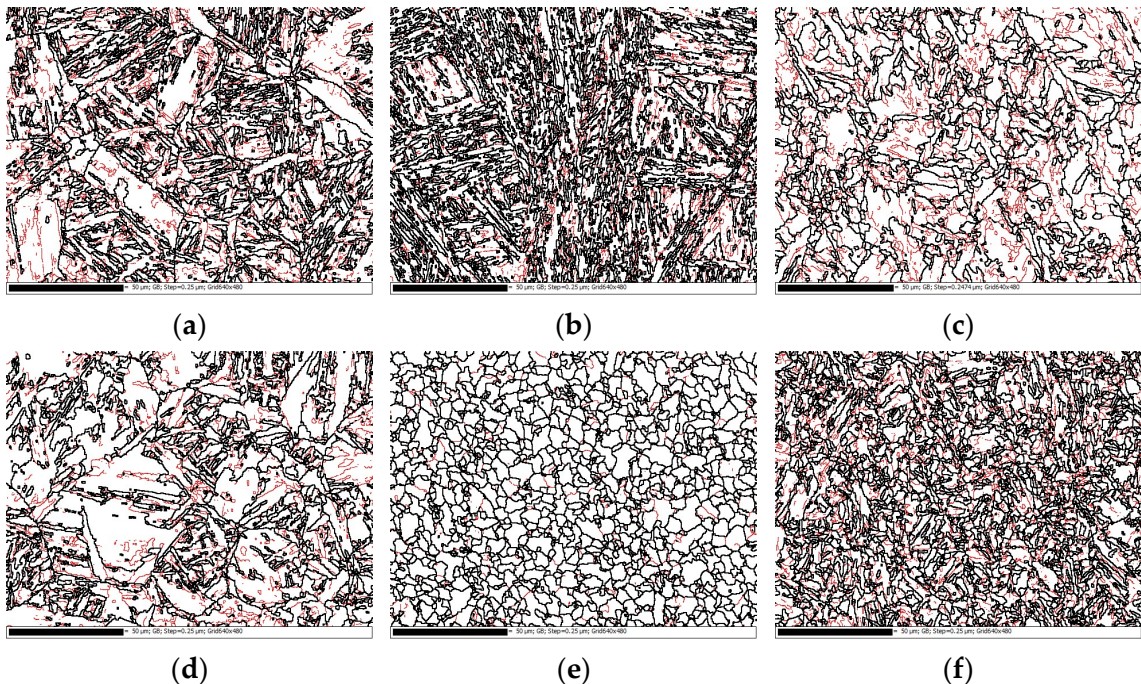

**Figure 12.** Distribution of HAGBs and LAGBs in the welding thermal simulation samples: (**a**) CGHAZ of 0 V; (**b**) GRHAZ of 0 V; (**c**) ICHAZ of 0 V; (**d**) CGHAZ of 10 V; (**e**) GRHAZ of 10 V; and (**f**) ICHAZ of 10 V.

As shown in Figures 12 and 13, the grain boundaries that had a misorientation greater than 15° were characterized as high angle grain boundaries (HAGBs) and marked as black lines, while the grain boundaries that had a misorientation less than or equal to 15° were characterized as low angle grain boundaries (LAGBs) and marked as red lines. The simulated CGHAZ of the 0 V and 10 V steels exhibited a similar frequency of LAGBs, but the misorientation of HAGBs was concentrated in the range of 45–60° for the 0 V steel, while it was concentrated in the range of 45–58° for 10 V steel. In the simulated GRHAZ of the 0 V and 10 V steels, the frequency of LAGBs was less and the frequency of HAGBs was similar, but the misorientation of the 0 V steel was concentrated in the range of 50–60°, while that of thee 10 V steel was concentrated in the range of 15–60°. The frequency of LAGBs in the simulated ICHAZ of the 0 V and 10 V steels was similar, the misorientation of HAGBs of 0 V steel was concentrated in the range of 15–25° and 45–60°, while that of the 10 V steel was concentrated in the range of 15–25° and 40–60°.

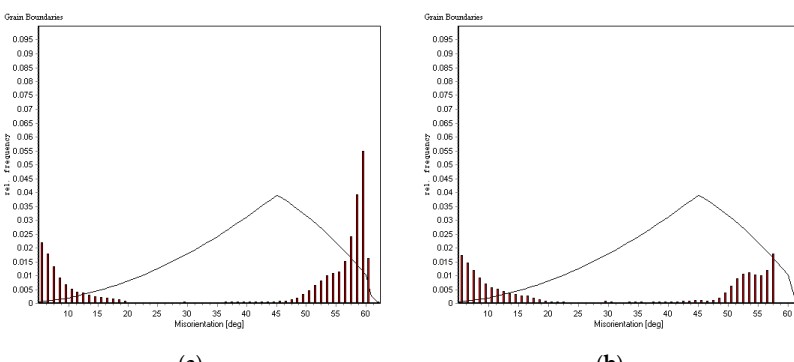

**Figure 13.** *Cont.*

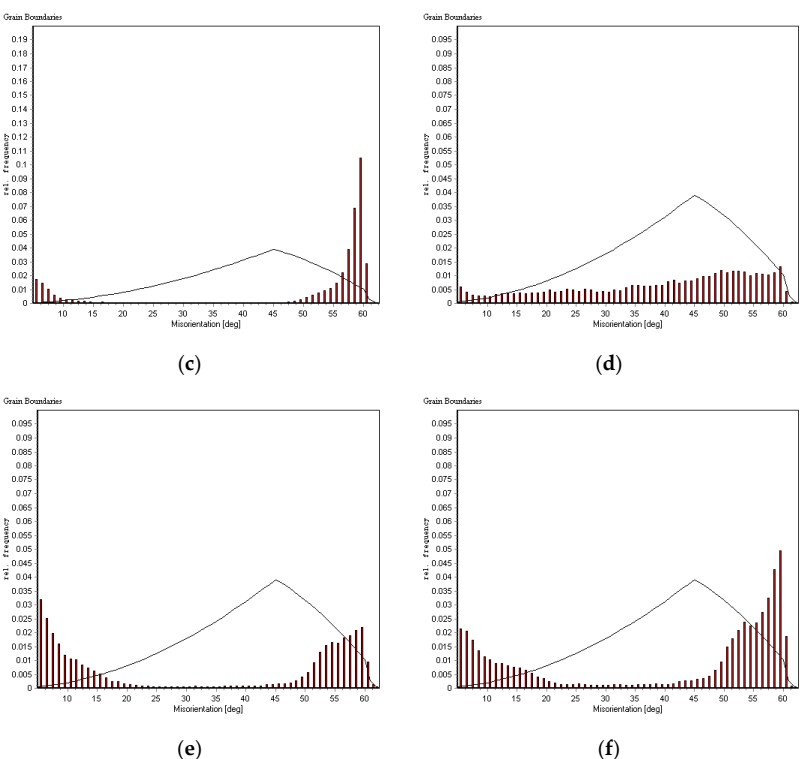

**Figure 13.** Frequency of HAGBs and LAGBs in the welding thermal simulation samples: (**a**) CGHAZ of 0 V; (**b**) CGHAZ of 10 V; (**c**) GRHAZ of 0 V; (**d**) GRHAZ of 10 V; (**e**) ICHAZ of 0 V; and (**f**) ICHAZ of 10 V.

The precipitation features in the welding thermal simulation samples of the 10 V steel were analyzed by TEM, and the distribution and elemental analysis of the precipitates are shown in Figure 14. In the simulated CGHAZ, almost no precipitates were observed; in the simulated GRHAZ, a small amount of precipitates with a small size were observed at the grain boundaries; and in the simulated ICHAZ, the micro-sized precipitates were found at the selected region.

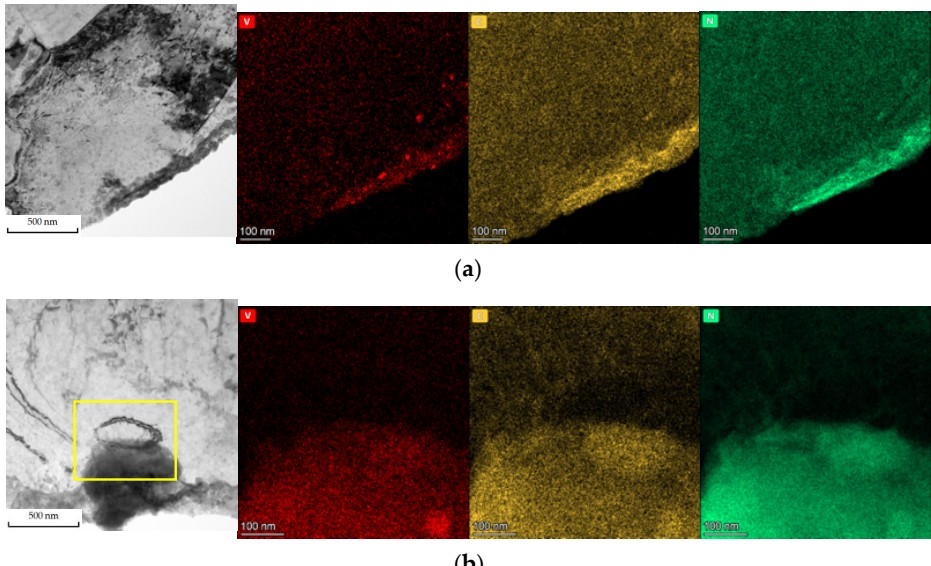

**Figure 14.** V(C, N) precipitates in the welding thermal simulation samples of the 10 V steel: (**a**) distribution and EDS analysis of precipitates in GRHAZ; and (**b**) distribution and EDS analysis of precipitates in ICHAZ.

## 4. Discussion

According to the testing results of V precipitates in the base material, the $V_f$ of V precipitates in the base material of 10 V steel was 0.196%, in which the $V_f$ of precipitates with a small size (2–<18 nm) was 0.145%, while the $V_f$ of precipitates with a large size (18–50 nm) was 0.051%. The peak values of the size distribution of V precipitates were in the range of 2–10 nm and 18–30 nm, respectively. However, the HAZ will experience a welding thermal cycle in the welding process, which may cause re-dissolution, re-precipitation, and growth of the V precipitates, thereby giving rise to differences in the microstructure and property of HAZ between the 0 V and 10 V steels.

### 4.1. Thermodynamic Simulation Calculation of V(C, N) Precipitation

Through thermodynamic simulation calculation, the dissolution and precipitation of V(C, N) in 10 V steel can be analyzed. The equilibrium solubility product equations of VC and VN in austenite are:

$$lg\{[V][C]\} = 6.72 - \frac{9500}{T} \tag{3}$$

$$lg\{[V][N]\} = 3.63 - \frac{8700}{T} \tag{4}$$

where [V], [C], [N] are the content of solid solution V, C, N elements (%), and T is the solution temperature in equilibrium (K).

According to the thermodynamic calculation method of the equilibrium solution of the ternary second phase illustrated by Yong Q. L. [26], the chemical equation of the ternary second phase V(C, N) generated by complete mutual dissolution of VC and VN can be indicated as $VC_xN_{1-x}$, and the chemical reaction is indicated as:

$$VC_xN_{1-x} = [V] + x[C] + (1 - x)[N] \tag{5}$$

Meanwhile, the content of V, C, and N elements in $VC_xN_{1-x}$ should realize the stoichiometric ratio, therefore, thee following equations can be obtained:

$$lg\frac{[V][C]}{x} = 6.72 - \frac{9500}{T} \tag{6}$$

$$lg\frac{[V][N]}{1-x} = 3.63 - \frac{8700}{T} \tag{7}$$

$$\frac{V - [V]}{C - [C]} = \frac{A_V}{xA_C} = \frac{50.9415}{12.0107x} \tag{8}$$

$$\frac{V - [V]}{N - [N]} = \frac{A_V}{(1-x)A_N} = \frac{50.9415}{14.0067(1-x)} \tag{9}$$

where $A_V$, $A_C$, $A_N$ are the relative atomic mass of the V, C, N elements, respectively, and V, C, N are the contents of V, C, N elements in 10 V steel, respectively (%).

When [V] = V, [C] = C, [N] = N, V(C, N) dissolves in steel completely, and the complete solution equation of V(C, N) can be obtained:

$$\frac{V \times C}{10^{6.72 - 9500/T_{AS}}} + \frac{V \times N}{10^{3.63 - 8700/T_{AS}}} = 1 \tag{10}$$

According to Equation (10), it can be calculated that the complete solution temperature $T_{AS}$ of V(C, N) in 10 V steel is 1352 K, namely 1079 °C.

According to Equations (6)–(9), equilibrium solid solution content [V], [C], [N] of the V, C, N elements and the value of x in $VC_xN_{1-x}$ can be obtained at a certain temperature T. As the solid solubility of VN in austenite is lower than VC, VN precipitates first at high

temperature, and the volume fraction of the ternary second phase V(C, N) that precipitates at temperature T in the equilibrium state can be obtained with Equation (11).

$$f = (V - [V]) \times \frac{A_{VCN}}{A_V} \times \frac{d_{Fe}}{100d_{VCN}} \tag{11}$$

where $A_{VCN} = xA_{VC} + (1 - x)A_{VN}$, $d_{VCN} = xd_{VC} + (1 - x)d_{VN}$, $d_{Fe}$, $d_{VC}$, $d_{VN}$ are densities of Fe, VC, and VN, respectively.

Through thermodynamic simulation calculation, the volume fraction of $VC_xN_{1-x}$ precipitates in 10 V steel at various temperatures T in the equilibrium state can be obtained, and the volume fraction variation of $VC_xN_{1-x}$ with the temperature can be drawn, as shown in Figure 15. Combined with the welding thermal cycle conditions that various zones of the HAZ experience in the welding process, the transformation law of V(C, N) precipitates and the microstructure of the HAZ of 10 V steel can be analyzed.

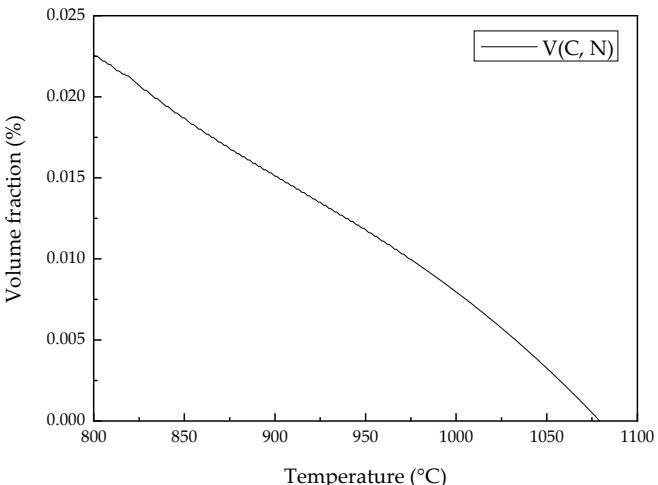

**Figure 15.** Volume fraction variation of V(C, N) in 10 V steel with temperature.

### 4.2. CGHAZ

The peak temperature of CGHAZ was high, and reached more than 1100 °C, and the peak temperature $T_p$ of the simulated CGHAZ in the welding thermal simulation experiment was 1300 °C, which is higher than 1079 °C, the complete solution temperature of V(C, N) in 10 V steel. Thus, V(C, N) in the CGHAZ of 10 V steel was completely dissolved in austenite in the welding heating process. In the subsequent cooling process, due to the fast cooling rate (the cooling time $t_{8/5}$ of simulated CGHAZ was 12 s) and the lack of large precipitates acting as nucleation sites, the time and power of the re-precipitation of V(C, N) were insufficient, therefore, the V and N elements in CGHAZ mainly dissolved in steel, which is the reason why even a small amount of precipitate was difficult to observe in the simulated CGHAZ. The microstructure of the real weld HAZ and simulated HAZ of 0 V and 10 V steels was similar, and consisted of lath bainite, granular bainite, and a small amount of polygonal ferrite. However, according to the IPF maps of the simulated CGHAZ (Figure 11a,d), the distribution (Figure 12a,d), and frequency (Figure 13a,d) of HAGBs and LAGBs in the simulated CGHAZ, the PAG size of the 10 V steel was larger and the HAGB frequency of the 10 V steel decreased significantly. Based on the above analysis, the V and N elements in the CGHAZ of 10 V steel mainly dissolved in steel. The dissolved V in CGHAZ will decrease the finishing temperature of austenite transition $A_{c3}$ in the welding heating process, however, considering that the V:N of 10 V steel reached 6.45, which is far more than the VN stoichiometric ratio of 3.64, the content of dissolved N in CGHAZ was high, which will lead to the finishing temperature of austenite transition $A_{c3}$ increasing significantly in the welding heating process, thereby facilitating the growth of PAGs in the CGHAZ of 10 V steel [18]. Coarse PAGs will decrease the grain boundary area of PAGs,

thereby reducing the HAGB frequency. As the propagation of microcracks at HAGBs needs more energy, HAGBs will inhibit microcracks growing to a critical Griffith crack length to spread across grains [25], therefore, the coarse PAGs and less HAGB frequency in the simulated CGHAZ of 10 V steel resulted in the significant reduction in low temperature impact toughness.

*4.3. GRHAZ*

Peak temperature of GRHAZ was between $A_{c3}$ and 1100 °C (around 850–1100 °C), and the peak temperature $T_p$ of the simulated GRHAZ in the welding thermal simulation experiment was 1000 °C, which almost reached 1079 °C, the complete solution temperature of V(C, N) in the 10 V steel in the thermodynamic equilibrium state; moreover, the V(C, N) precipitates with a small size (2–<18 nm) accounted for around 74.0% of the total V(C, N) precipitates in the 10 V steel, and the re-dissolution of precipitates is more likely to take place with the precipitate size decreasing, therefore, V(C, N) in the simulated GRHAZ of 10 V steel was almost all dissolved in austenite in the welding heating process. In the following cooling process, VN precipitated first due to the smaller solid solubility in austenite, then VC precipitated, taking VN as the nucleation site to form V(C, N) particles with a decrease in temperature. Due to the fast cooling rate (the cooling time $t_{8/5}$ of simulated GRHAZ was 13 s), V(C, N) mainly precipitated at the grain boundary in small sizes (Figure 14a), and part of V(C, N) still dissolved in steel. Considering that the V:N of 10 V steel reached 6.45, which is far more than the VN stoichiometric ratio of 3.64, the content of dissolved N in the simulated GRHAZ of 10 V steel was high, which will lead to the start temperature of austenite transition $A_{r3}$ increasing in the cooling process; moreover, VN showed more potency as the nucleation site due to the low lattice mismatch with ferrite [27–29], so the austenite in the simulated GRHAZ of 10 V steel transformed to proeutectoid ferrite at high temperature in the welding cooling process. The excessive C atoms diffused into residual austenite to increase the stability, thus the residual austenite transformed to pearlite and martensite in the following cooling process. Due to the strong pinning effect on grain boundaries resulting from the precipitation of V(C, N) at the grain boundaries, the ferrite grain was refined. Therefore, in the condition of 1000 °C of peak temperature $T_p$ and 13 s of cooling time $t_{8/5}$, the microstructure of thee simulated GRHAZ of 0 V steel consisted of lath bainite, granular bainite, and polygonal ferrite, and the IPF map (Figure 11b) showed lath PAGs, while the microstructure of the simulated GRHAZ of 10 V steel consisted of refined polygonal ferrite, pearlite, and martensite, and the IPF map (Figure 11e) showed refined polygonal ferrite grains. The difference in the metallographic structure of the simulated GRHAZ between 0 V and 10 V steels caused the difference in HAGBs and LAGBs. According to the distribution (Figure 12b,e) and frequency (Figure 13b,e) of HAGBs and LAGBs in the simulated GRHAZ, the frequency of HAGBs in the simulated GRHAZ of 0 V and 10 V steels was similar, but the misorientation of HAGBs in the simulated GRHAZ of the 0 V steel was concentrated in the range of 50–60°, while the misorientation of HAGBs in the simulated GRHAZ of 10 V steel was concentrated in the range of 15–60°, so the inhibition effect of HAGBs in the simulated GRHAZ of 10 V steel on crack propagation decreased, leading to the reduction in impact toughness of the simulated GRHAZ of 10 V steel.

However, in the real weld HAZ, peak temperature decreased and cooling time increased with the increase in distance to the fusion line to cause temperature gradient, while there was not a great temperature gradient in the simulated HAZ. The simulated HAZ sample only simulated the welding thermal cycle at one point in various zones of real weld HAZ, thus the simulated HAZ sample only characterized the microstructure and property at this point. Therefore, there were differences in the microstructure and property between the real weld GRHAZ and simulated GRHAZ. In real weld GRHAZ, peak temperature decreased and cooling time increased with the increase in distance to fusion line, which will cause the cooling rate to decrease to move into thee ferrite phase transition zone in the welding cooling process, thereby reducing the difference in microstructure evolution

between the 0 V and 10 V steels caused by V–N microalloying in the weld thermal cycle. Moreover, with the increase in distance to fusion line, the peak temperature decreases and cooling time increases to produce V(C, N) precipitates in real weld GRHAZ that are difficult to completely dissolve and have sufficient time to experience re-precipitation, and growth through the Ostwald ripening effect [30]. As V(C, N) precipitates with large sizes will break the continuity of the steel matrix and act as crack initiation points, the impact toughness of the real weld GRHAZ of 10 V steel decreases.

*4.4. ICHAZ*

Peak temperature of ICHAZ was between $A_{c1}$ and $A_{c3}$ (around 700–850 °C), the cooling rate of ICHAZ was relatively low, and the peak temperature $T_p$ and cooling time $t_{8/5}$ of the simulated ICHAZ was 830 °C and 17 s, respectively. Therefore, in ICHAZ, part of the microstructure transformed to austenite and part of the ferrite reserved and grew in the welding heating process, then the austenite transformed to ferrite, pearlite, and martensite in the following cooling process. As the grain refinement effect resulted from the precipitation of V(C, N), the grain size of the base material of 10 V steel was small, the microstructure of the ICHAZ of 10 V steel was refined, which was proven by the microstructure measurement of the real weld ICHAZ (Figure 7g,h), simulated ICHAZ (Figure 10e,f), and IPF maps of simulated ICHAZ (Figure 11c,f). Moreover, according to the distribution (Figure 12c,f) and frequency (Figure 13c,f) of HAGBs and LAGBs in the simulated ICHAZ, the size of the ferrite grains in the simulated ICHAZ of 10 V steel was smaller, so the frequency of HAGBs in the simulated ICHAZ of 10 V steel was higher, which is beneficial in improving the impact toughness. However, due to the relatively low cooling rate, V(C, N) in the simulated ICHAZ of 10 V steel had sufficient time to experience re-dissolution, re-precipitation, and growth through the Ostwald ripening effect, so it is difficult to find V(C, N) precipitates with a 2–50 nm diameter, but V(C, N) precipitates with a 5 μm of diameter were observed. As V(C, N) precipitates with a micron size will break the continuity of the steel matrix and act as crack initiation points, the impact toughness of the simulated ICHAZ of 10 V steel deteriorated. Therefore, although the frequency of HAGBs in the simulated ICHAZ of 10 V steel was relatively high to improve the impact toughness, V(C, N) precipitates of a micron size deteriorated the low temperature impact toughness, thereby leading to the reduction in the low temperature impact toughness of the simulated ICHAZ of 10 V steel.

**5. Conclusions**

1. V and N elements in the CGHAZ of steel containing 0.10% V and 0.0155% N mainly dissolved in steel, considering that the V:N was far more than the VN stoichiometric ratio of 3.64, so the content of dissolved N in the CGHAZ was high, which will lead to the $A_{c3}$ temperature increasing significantly to facilitate the growth of PAGs. Coarse PAGs will decrease the grain boundary area of PAGs to reduce HAGB frequency, thereby leading to the significant reduction in the low temperature impact toughness of the simulated CGHAZ of steel containing 0.10% V and 0.0155% N, specifically, the mean −20 °C impact value decreased from 87 J to 18 J.

2. With the addition of 0.10% V and 0.0155% N into steel, the microstructure of simulated GRHAZ transformed from lath bainite, granular bainite, and a small amount of ferrite to polygonal ferrite, pearlite, and martensite, and the misorientation of HAGBs decreased, causing the lower impact toughness and decreased microhardness of the simulated GRHAZ of steel, specifically, the mean microhardness decreased from 335 HV0.2 to 207 HV0.2, and the mean −20 °C impact value decreased from 117 J to 103 J. However, in the real weld GRHAZ, peak temperature decreased and cooling time prolonged due to an increase in the distance to fusion line, therefore reducing the difference in the microstructure evolution caused by V–N microalloying.

3. The microstructure of the ICHAZ of steel containing 0.10% V and 0.0155% N was refined to increase the frequency of HAGBs, which is beneficial in improving the

impact toughness. However, V(C, N) in the simulated ICHAZ had sufficient time to experience re-dissolution, re-precipitation, and growth to form micro-sized V(C, N) precipitates, which will deteriorate the low temperature impact toughness, specifically, the mean $-20\,^\circ$C impact value decreased from 93 J to 62 J.

**Author Contributions:** Conceptualization, S.Y., H.Z. and G.W.; Methodology, investigation, K.C., Z.L. and H.Y.; Writing, K.C.; Writing, review and editing, S.Y. and X.N.; Supervision, S.Y. and G.W. All authors have read and agreed to the published version of the manuscript.

**Funding:** This research was funded by the Natural Science Foundation of Shandong Province (grant number ZR2020ME146), the Natural Science Foundation of China (grant number U1937205), and the Major Scientific and Technological Innovation Projects in Shandong Province (grant numbers 2020CXGC010303, 2019TSLH0103). The APC was funded by the Natural Science Foundation of Shandong Province (grant number ZR2020ME146).

**Institutional Review Board Statement:** Not applicable.

**Informed Consent Statement:** Not applicable.

**Data Availability Statement:** The study did not report any data.

**Conflicts of Interest:** The authors declare no conflict of interests.

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
