# Peer review of "Effects of V–N Microalloying on Microstructure and Property in the Welding Heat Affected Zone of Constructional Steel"

_metals, doi:10.3390/met12030480_

Round 1
Reviewer 1 Report
1/ Selection of simulated thermal cycles (Table 3) is questionable and wants justification in what concerns cooling rates. On the one hand, when ignoring the plate thickness effect, a more distance from the weld metal i.e. a lower peak temperature should decelerate cooling indeed. On the other hand, in the considered case (Fig.5) width of the HAZ is comparable with rather small thickness of the plate so that the cooling of more distant areas (GRHAZ and ICHAZ) is stronger accelerated by surfaces. Even avoiding a numerical solution of the thermal boundary problem, the authors should clarify this issue. This gains in significance as they finally recognize irrelevance of the simulated cycles to REAL conditions. Meanwhile, the mentioned origin of such an error in some uncertain “temperature gradient” (last paragraph of subsection 4.3) is not clear enough.
2/ Misprinted “1.0V ” in Abstract should be substituted by “0.1%V”. Even if the authors have meant “10V” according to Table 1, this is untimely before this denotation appears in the paper body.
Author Response
Dear reviewer and editor,
Thank you very much for your attention and the referee’s evaluation and comments on our paper.
We have revised the manuscript according to your kind advices and referee’s detailed suggestions. Enclosed please find the responses to the referee. We sincerely hope this manuscript will be finally acceptable to be published on Metals.
Thank you very much for all your help and looking forward to hearing from you soon.
Best regards,
Sincerely yours
Shengjie Yao

Reviewer 2 Report
Dear Authors,
I have read your paper "Effects of V-N micro-alloying on microstructure and property in the welding heat affected zone of constructional steel".
It fulfills the aims and scope of the Metals journal. Presented investigations are interesting. My comments and suggestions are listed below.
General remarks:
- I propose to add the quantitative results into the abstract. Moroever, please mention, which tests were performed.
- Pages 17-18 - please fill the red areas with relevant information.
Introduction:
- This section is quite short. However, you have described relevant topic in details. I have no suggestions here.
Materials and Methods:
- Table 1 - please show the carbon equivalent values of both steeld. This factor allows to assessment the weldability of used materials.
- Not "WS-300 DC welder" but "welding power source".
- Why presented welding parameters were used? Preeliminary investigations/filler material manufacturer data/literature proceedings? Please add relevant info.
- Subsection 2.2.3. - you have described parameters of used tests. Have they were taken in accordance with relevant standards' requirements? If yes, please show relevant standard numbers. If not, please describe why.
Results:
- I propose change "base metal" to "base material", as it is commonly used in welding engineering standards.
- Fig. 2 - please show bigger scale bars.
- Table 4 - how many measurements were taken to get these values? Any standard deviation values?
- Fig. 3 - scale bars should be improved.
- 3.2 - you presented more than one property, so the title of subsection should be "properties".
- 3.3. - please show standard deviation values.
- Fig. 10 - their size is unacceptable. Please improve. The same with FIg. 13.
Discussion:
- This section is really well written. You have diuscussed presented results. Moreover, results were compared to other scientific papers. Congratulations, realy good work.
Conclusions:WS-300 DC
- Please support conclusions with the quantitative results.
Author Response

(The authors gave the same response as above.)
